Real-life experience with personally familiar faces enhances discrimination based on global information

Ramon Meike 1 2 meike.ramon@gmail.com meike.ramon@unifr.ch
Van Belle Goedele 1
1 Institute of Research in Psychology, Institute of Neuroscience, Université Catholique de Louvain , Louvain-La-Neuve , Belgium
2 Department of Psychology, University of Fribourg , Fribourg , Switzerland
Abdullah Jafri
Electronic publication date: 2016 Jan 4
Publication date: 2016
Volume: 4
Electronic Location ID: e1465
Received 2015 Jul 2; Accepted 2015 Nov 11
Copyright: ©2016 Ramon and Van Belle
Copyright year: 2016
Copyright holder: Ramon and Van Belle
License: This is an open access article distributed under the terms of the Creative Commons Attribution License, which permits unrestricted use, distribution, reproduction and adaptation in any medium and for any purpose provided that it is properly attributed. For attribution, the original author(s), title, publication source (PeerJ) and either DOI or URL of the article must be cited.
License URL: https://creativecommons.org/licenses/by/4.0/

Keywords: Face processing, Global information integration, Holistic processing, Real-life exposure, Personal familiarity

Funding: Belgian National Foundation for Scientific Research (FNRS) This work was supported through funding by the Belgian National Foundation for Scientific Research (FNRS). The funders had no role in study design, data collection and analysis, decision to publish, or preparation of the manuscript.

==============================
Despite the agreement that experience with faces leads to more efficient processing, the underlying mechanisms remain largely unknown. Building on empirical evidence from unfamiliar face processing in healthy populations and neuropsychological patients, the present experiment tested the hypothesis that personal familiarity is associated with superior discrimination when identity information is derived based on global, as opposed to local facial information. Diagnosticity and availability of local and global information was manipulated through varied physical similarity and spatial resolution of morph faces created from personally familiar or unfamiliar faces. We found that discrimination of subtle changes between highly similar morph faces was unaffected by familiarity. Contrariwise, relatively more pronounced physical (i.e., identity) differences were more efficiently discriminated for personally familiar faces, indicating more efficient processing of global, as opposed to local facial information through real-life experience.

Introduction

Humans are highly efficient at processing faces of conspecifics. Within a few hundred milliseconds, we can categorize faces according to their gender, expression, race and familiarity, as well as identify them. The social importance of face processing, and its complexity as a perceptual and cognitive process has motivated numerous investigations of the underlying mechanisms. Several lines of research indicate that processing of identity information is linked to observers’ ability to integrate global facial information, also referred to as holistic processing, a hallmark of adult’s face processing expertise (McKone, Kanwisher & Duchaine, 2007; Mondloch et al., 2007; Richler & Gauthier, 2013). Neuropsychological studies have demonstrated that face processing deficits observed in prosopagnosia can be accounted for by patients’ impairment of holistic processing (e.g., Ramon, Busigny & Rossion, 2010; for a review see Rossion, 2014). Furthermore, recent evidence suggests a direct relation between the extent of holistic processing and healthy observers’ face processing efficiency (Wang et al., 2012).

While the most impressive demonstration of humans’ face processing ability is observed for familiar faces, the bulk of empirical evidence regarding its determinants stems from experiments involving unfamiliar face stimuli. Several studies indicate that personally familiar faces are processed more efficiently than their unfamiliar counterparts (e.g., Bruce et al., 1999; Burton et al., 1999; Ramon, Caharel & Rossion, 2011). Familiar identities can be behaviorally categorized as such significantly faster than unfamiliar faces, within as little as 310 ms (Ramon, Caharel & Rossion, 2011; see also Busigny et al., 2012), with differential electrophysiological responses occurring 100 ms prior (Caharel, Ramon & Rossion, 2014). Moreover, the presence of an underlying facial representation stored in memory makes perceptual processing more robust to variations in the input characteristics. For instance, we can easily recognize a famous or personally familiar face based on their eyes alone (Sadr, Jarudi & Sinha, 2003; Ramon et al., 2015), or from a highly degraded image (Watier & Collin, 2009; Loftus & Harley, 2005; Ramon et al., 2015), and even after considerable time periods (Bahrick, Bahrick & Wittingler, 1975). Contrariwise, identity processing in unfamiliar faces is highly dependent on the visual characteristics of the stimulus input, i.e., availability of color, texture, and surface reflectance (Russell et al., 2006; Jiang, Blanz & Rossion, 2011), and is hence more prone to error given superficial image variations (e.g., viewpoint or image quality; Bruce, 1986; Roberts & Bruce, 1989; Bruce et al., 2001).

Together, these observations support the general consensus that unfamiliar and personally familiar faces are processed differently (cf., Tong & Nakayama, 1999; Megreya & Burton, 2006; Carbon, 2008; Gobbini et al., 2013). However, there is little understanding of the underlying mechanisms promoting such differential processing. Early studies using famous faces have suggested that familiarity affects processing of internal facial information in general (Ellis, Shepherd & Davies, 1979; Young et al., 1985; Brooks & Kemp, 2007). Other investigations have provided inconsistent results regarding whether or not the effects of familiarity are restricted to processing of the eyes (O’Donnell & Bruce, 2001), or extend to the less salient mouth region (Barton et al., 2006; Van Belle et al., 2010d). However, more recent findings may reconcile these seemingly conflicting findings. Ramon (2015a); Ramon, (2015b) reported that personal familiarity affects discrimination of vertical displacements of the eyes and the mouth, as well as changes of the overall configuration between these two sources of information. These findings suggest that personal familiarity may not affect processing of specific types of facial information, but rather modulate the processing style engaged in.

The present study sought to extend these previous findings by varying the degree to which local or global information was diagnostic for identity discrimination. The underlying idea is that personal familiarity facilitates perceptual processing—and thus discrimination—of faces through the presence of a facial representation in memory. Given the relationship between holistic processing and processing of identity (Sergent, 1984; Tanaka & Farah, 1993; Tanaka & Farah, 2003), our hypothesis was that such enhanced perceptual processing for familiar compared to unfamiliar faces would be observed along with decreased reliance on local, piecemeal information.

In a delayed matching paradigm, observers performed forced-choice decisions of facial identity. The face stimuli used to this end were derived from morph continua similar to those used in investigations of categorical perception of identity, or expression (Beale & Keil, 1995; Gilaie-Dotan & Malach, 2007; Fox et al., 2009; Ramon, Dricot & Rossion, 2010). The parametric variations in identity-related physical information offered a means to manipulate the perceptual similarity and hence ambiguity of information supporting discriminative decisions.1While dissimilar stimuli could be more easily discriminated based on global information, discrimination of more similar ones would be comparably less efficient and require more local distinctive feature sampling. Hence, our experimental conditions differed in the extent to which discrimination would be based on global, or holistic processing (defined here as the fast and automatic process leading to an internal representation of the face as a whole; Galton, 1883; Rossion, 2013).

Importantly, to investigate whether personal familiarity selectively affects global processing, or leads to a general processing advantage, two groups of observers completed the same task. Control subjects were unfamiliar with all of the identities depicted. The experimental group comprised subjects who were highly personally familiar with half of the identities, which represented their classmates.

Thus, contrary to previous studies, we varied two aspects that are considered to modulate observers’ face processing efficiency. On the one hand, the physical similarity of face stimuli (which co-varies with discrimination efficiency based on global information), as well as observers’ familiarity with face stimuli. In line with the high discrimination performance reported for familiar vs. unfamiliar face discrimination, a beneficial effect of familiarity was anticipated for conditions of low similarity considered to involve reliance on global information. Contrariwise, no such familiarity-related advantage was expected for discrimination of highly similar faces, which would rely on observers’ ability to identify locally circumscribed details.

Naturally, local details are also available in conditions of low similarity, and thus could be used for face discrimination. Therefore, we incorporated a third condition involving low similarity. Here, the local, high resolution information was made unavailable through stimulus blurring (Sergent, 1986; Collishaw & Hole, 2000; Schwaninger et al., 2006; Gilad-Gutnick, Yovel & Sinha, 2012). Thus, we removed high spatial frequency information typically used for piecemeal analytic processing (Goffaux, 2009; Goffaux & Rossion, 2006).

In sum, we manipulated stimulus similarity, availability of high resolution local details, as well as personal familiarity to directly test—for the first time to our knowledge—the hypothesis that familiarity leads to an advantage in global/holistic, but not local processing. Two possible outcomes were anticipated. First, if familiarity is associated with an advantage at processing local discriminative information, we should observe a familiarity advantage across both conditions involving high resolution images enabling the use of said information. Alternatively, if familiarity is advantageous for automatic global processing, we should observe an experience-related benefit for conditions involving low stimulus similarity—regardless of whether high spatial information used for local information processing is available.

Methods

Procedure and apparatus

Participants performed a two-to-one alternative forced-choice delayed matching task. Each trial started with a centrally presented fixation cross. Upon its fixation, two probe faces (distance between the inner borders: 5° of visual angle) were presented side by side for 2.5 s, during which they could be explored freely. Immediately thereafter the probe faces were replaced by a single centrally presented test face, which was identical to one of the two probes (sides counterbalanced). The test face remained on screen until participants indicated to which of the probes it corresponded by pressing the corresponding left or right arrow button on the keyboard. The next trial was initiated immediately after each response.

The experiment consisted of six blocks, with each block comprising 40 trials; three blocks involved presentation of familiar and unfamiliar faces, respectively. Each block consisted of trials of one of the three experimental conditions (Full 20%, Full 50%, Blur 50%; see below). The order of the blocks in terms of familiarity and experimental condition was counterbalanced across participants (note that control subjects were unfamiliar with all faces used as stimuli; see below). Within each block there was an equal amount of left and right correct response sides and both faces from each of the 20 stimulus pairs appeared twice as test stimuli. This procedure ensured equal likelihood of perceiving either of the identities of a given morph continuum (see Ramon, Dricot & Rossion, 2010).

To become familiar with the task, participants completed five practice trials prior to the main experiment. These practice trials contained faces that were not used in the main experiment, and were excluded from the analyses. Stimuli were displayed using Presentation or Eprime software, on a 22” Sony Trinitron monitor (58 cm viewing distance, 1,400 × 1,050 pixel resolution, 85 Hz refresh rate). Probe and test faces’ height comprised 10.3 and 11.5° of visual angle, respectively. This roughly corresponds to the size of a real face viewed at normal conversational distance of 90 cm (Hall, 1966). Both stimulus display and response registration was controlled by an Intel Centrino vPro.

Stimuli

Two different sets of stimuli were used in the main experiment. The first was taken from a previous study (Ramon, Dricot & Rossion, 2010) and involved 10 morph continua, the extremes of which were unfamiliar to all participants. An additional set of 10 morph continua were created between pairs of faces with which half of the participants were personally familiar: their classmates. To this end, full-front photographs of 26 students of the same class were taken under identical conditions (distance, lighting). The photographs of five male and one female student were excluded from the final set of (Caucasian, female) familiar faces used, due to the presence of facial hair or make-up at the time the photographs were taken. Using Adobe Photoshop, the remaining 20 familiar face stimuli were cropped of external features and hair (see Fig. 1A) and morph continua were created with Photo Morpher v3.10 (Morpheus, Santa Barbara, CA, USA) following the same procedure as used by Ramon, Dricot & Rossion (2010). Specifically, face pairs were selected based on eye color, shape and overall luminosity (average pixel intensity) of the face. For each face, 350 points were placed on the critical features (encompassing the pupils, iris, eye bulbs, eye lids, eye brows, mouth, nose and overall facial contour) to allow smooth transitions between the stimuli created per morph continuum (two original faces representing the extremes, with 10% increments; see Fig. 1A).

Figure 1 Stimuli discriminated in the delayed matching task.

(A) An example of a familiar face morph continuum. Unfamiliar stimuli were those created and used in a previous study (see Ramon, Dricot & Rossion, 2010 for examples and details regarding stimulus creation), which were unfamiliar to all subjects tested. Familiar morph continua were created from pairs of classmates of experimental subjects tested, and were unfamiliar to control subjects. (B) Examples of stimulus pairs to be discriminated in the 2AFC delayed matching task (pairs were taken from either side of a continuum).

As described above, our hypothesis was that personal familiarity is associated with facilitated face discrimination, the degree of which would depend on the physical similarity of the identities discriminated. Therefore, we created three conditions which differed in terms of their reliance on local distinctive features, or in other words, the extent to which performance is determined by global processing. The first condition involved probe stimuli, which differed by 20% (i.e., physically similar pairs, Full 20%) and were located on the same side of the categorical boundary of the morph continua (i.e., the point where both identities would be perceived with equal likelihood; see e.g., Beale & Keil, 1995; Gilaie-Dotan & Malach, 2007; Ramon, Dricot & Rossion, 2010; Rotshtein et al., 2005; note that as here we were not interested in testing categorical perception, but rather the effects of stimulus similarity, the categorical boundary was considered as the midpoint of the morph continua and was hence not individually defined). The second condition involved pairs of stimuli, which differed by 50% (i.e., physically dissimilar pairs, Full 50%) and were located on opposite sides of the categorical boundary of the morph continua. Note that the distance of these more dissimilar items relative to both extremes (i.e., original faces), and the categorical boundary was identical to the physical difference between Full 20% pairs (see Fig. 1).2In the Full 20% condition the faces are more similar than in the Full 50% condition, making the comparison more ambiguous, and comparably more dependent on local information (see e.g., Barton et al., 2006). A third condition involved the same dissimilar pairs (i.e., those used in the Full 50% conditions), to which a Gaussian blur (30 pixel radius; see e.g., Gilad-Gutnick, Yovel & Sinha, 2012) was applied to make high-resolution information (e.g., freckles, wrinkles, etc.) unavailable. These physically dissimilar pairs of blurred faces (Blur 50%) comprised the third experimental condition, created to disable to the use of local features in the low similarity condition.

Participants

Twelve participants (mean age: 23 ± 1; 3 male), who were personally familiar with half of the individuals depicted in the stimuli (from here on referred to as participants comprising the ‘experimental group’), were financially compensated for their participation. They were all senior year psychology master students who had been following classes in the same group of ∼30 students for about two years at the time of testing; some knew each other for a maximum of 5 years (data collection occurred while students were still in the cohort). The control group comprised 12 participants (mean age: 25 ± 4; 4 male), who were unfamiliar with all individuals’ images used to create the face stimuli, and were also financially compensated for their participation. The experiments were undertaken with the understanding and written consent of each subject, and conform to The Code of Ethics of the World Medical Association (Declaration of Helsinki).

Analyses and results

The analyses were conducted separately on accuracy scores and correct RTs as individual subjects may differ in terms of the measure they exhibit performance differences (Pachella, 1974; Meyer et al., 1988). Raw accuracy and RT values per group and condition, as well as 95% bootstrapped confidence intervals can be found in Table S1. None of the conditions were associated with chance, or ceiling level performance; outliers (trials in which the RT exceeded the average RT ±2SD per condition and subject) were removed from the data.

To test the hypothesis that familiarity affects performance differently across conditions, Generalized Estimating Equations were applied to test for a three way interaction in a repeated measures model with group (experimental vs. control) as a between-subjects factor, and condition (Full 50%, Full 20%, Blur 50%) and familiarity of stimuli (familiar vs. unfamiliar) as within-subjects factors. Using a binomial logit link distribution for accuracy, and a normal distribution for RT, we observed a significant three-way interaction for accuracy scores (Wald Chi2(2) = 13.82, p < .01), but not for correct RTs (Wald Chi2(2) = 3.36, p = .19).

To further investigate this interaction, we performed Bonferroni corrected post-hoc contrasts between individual factor level combinations. To facilitate the interpretation of these contrasts, they were performed on familiarity indices computed for each subject and condition ((familiar − unfamiliar) / (familiar + unfamiliar)). These familiarity indices capture potential effects of stimulus familiarity and will be referred to as ‘the familiarity advantage’ in the following. Group means of the familiarity advantage for accuracy across conditions are displayed in Fig. 2.

Figure 2 Familiarity advantage in the 2AFC delayed matching task with personally familiar and unfamiliar morph stimuli.

Mean familiarity advantage ((familiar − unfamiliar)/(familiar + unfamiliar)) for accuracy scores per condition observed for control, as well as experimental subjects. Error bars represent standard errors for both measures. Note that for control subjects, all faces presented were unfamiliar.

T-tests revealed that the familiarity advantage was larger for the experimental group than for the control group in the Full 50% condition (t = 2.80, p < .05) and in the Blur 50% condition (t = 2.55, p < .05), but not in the Full 20% condition (t = .64, p = .53). Furthermore, in the experimental group the familiarity advantage was significantly smaller in the Full 20% than in the Full 50% (t = 2.73, p < .05) condition, and even larger in the Blur 50% (t = 3.98, p < .01) than the Full 50% condition. In the control group, the difference between familiar and unfamiliar stimuli did not significantly vary across conditions (ps > .05).

Discussion

Several lines of empirical evidence suggest a relationship between face processing efficiency and the ability to rapidly integrate information from across the entire face into a unified percept, also referred to as holistic processing. Some studies indicate a direct association between the degree of holistic processing exhibited by healthy observers, and the efficiency with which upright faces are processed (e.g., Wang et al., 2012). Experimental manipulations utilized to disrupt holistic processing include stimulus inversion, as well as increased physical stimulus similarity. Both lead to reliable decreases in face processing performance and have been associated with employment of a more local/featural processing style (Barton et al., 2006; Orban de Xivry et al., 2008; Van Belle et al., 2010a), a phenomenon also observed in patients with prosopagnosia (i.e., the face selective recognition deficit due to brain damage), who have lost the ability to process faces holistically (e.g., Bukach et al., 2008; Busigny & Rossion, 2010; Ramon, Busigny & Rossion, 2010; Van Belle et al., 2011; Van Belle et al., 2010c; Rossion, 2014).

The present study aimed to investigate the effect of repeated, real-life experience with personally familiar individuals on perceptual processing of faces. Naturally, healthy observers have no difficulty in determining the identity of familiar individuals (see also e.g., Jenkins et al., 2011; Ramon et al., 2015)—a task for which ceiling effects can be expected. To manipulate the relative reliance on global versus local information processing for face discrimination, the stimulus material used here involved morph faces of varied physical similarity (e.g., Beale & Keil, 1995; Gilaie-Dotan & Malach, 2007; Ramon, Dricot & Rossion, 2010; Rotshtein et al., 2005). Following previous research, irrespective of familiarity, discrimination of highly similar faces was assumed to be less efficient and rely more on processing of local details (Barton et al., 2006; Orban de Xivry et al., 2008). Contrariwise, dissimilar faces were anticipated to be distinguished more efficiently given automatic extraction of global information from across the entire face.

Most importantly, the experimental subjects tested here were personally familiar with half of the individuals used to create the morph stimuli presented. That is, not only did the face stimuli differ in their respective physical similarity, but also regarding the presence of a facial representation stored in memory. Replicating previous findings (e.g., Bruce et al., 1999; Burton et al., 1999; Ramon, Caharel & Rossion, 2011; Ramon, 2015a; Ramon, 2015b), we found that personal familiarity was associated with enhanced face discrimination performance. Using morph stimuli differing in physical similarity and subjects’ familiarity, we sought to determine whether this enhancement is due to more efficient global, as opposed to local processing; two potential outcomes were anticipated. Increased performance for familiar versus unfamiliar faces for high resolution images only (i.e., irrespective of physical (dis)similarity) would indicate higher efficiency at discriminating faces based on local information. Alternatively, a familiarity-dependent advantage for discrimination of dissimilar faces only (i.e., for both high resolution and blurred images) would support the idea of experience-related increased efficiency for discerning facial identity changes based on global information.

The performance profiles observed for the discrimination of morph stimuli created from personally familiar faces was markedly different from that of unfamiliar faces. First, discrimination of highly similar (Full 20%) face morphs, which highly relies on local/featural processing (e.g., Barton et al., 2006; Orban de Xivry et al., 2008), was unaffected by familiarity. This finding, which cannot be accounted for in terms of floor effects, indicates that familiarity, i.e., extensive prior real-life experience, does not lead to more proficient performance when processing relies on local information. Note that this coincides with Barton et al’s (2006) findings that higher ambiguity leads to an increase in difficulty of perceptually based decisions, as well as the need to accumulate more data (in their study: more fixations, longer durations; see also Althoff & Cohen, 1999). Second, mirroring the high efficiency with which personally familiar faces are generally processed, performance increased when discrimination of high-resolution, dissimilar faces (Full 50%) was required. Moreover, performance at discriminating the same level of similarity was superior for familiar relative to unfamiliar stimuli despite the unavailability of high-resolution, local information (Blur 50%). These results are a clear indication that facial representations stored in memory, as available for personally familiar faces, facilitate global processing and reduce reliance on high-resolution local information for face discrimination.

Increased holistic processing of personally familiar faces?

To the extent that discrimination of the dissimilar conditions applied here (Full 50%, Blur 50%) could be considered to tap into holistic processing, the present findings would support the notion that visual experience modulates holistic processing (see e.g., Rossion, 2013). Previous studies exploring the effects of cross-cultural and cohort-dependent exposure have reported increased holistic processing and superior face discrimination for faces with which subjects had extensive exposure, e.g., own-race (Michel et al., 2006; Michel, Corneille & Rossion, 2007), own-age faces (De Heering & Rossion, 2008; Kuefner et al., 2008), and faces presented in their canonical orientation (Van Belle et al., 2010b). The present findings may therefore add to a body of evidence suggesting a direct relationship between experience, increased holistic processing and face processing efficiency (see also e.g., Crookes, Favelle & Hayward, 2013; Degutis et al., 2013; Susilo et al., 2009; Proietti, Pisacane & Macchi Cassia, 2013). Moreover, our results expand on these findings, as here a modulatory effect was related not to exposure with a specific category of faces (own age, same-race, upright), but specific exemplars of the same category.

Conclusion

To summarize, in keeping with the observation that personally familiar face identification is robust across viewing distances (Ramon, 2015b) and therefore efficient even provided only low spatial frequency information, we observed that familiarity was associated with a decreased reliance on local details for discrimination of facial identity. Experience-related facilitation of perceptual processing was found when global information was diagnostic for face discrimination (i.e., for dissimilar stimuli and when visual information was degraded). Our findings demonstrate that individual face representations obtained through real-life interactions and stored in memory enhance observers’ ability to discriminate identity-related information flexibly depending on the visual input available.

Supplemental Information

Table S1 Table S1 provides the accuracy scores and correct RTs per condition along with 95% bootstrapped confidence intervals.

These were obtained per condition and group by randomly sampling subjects with replacement; this process was repeated 999 times, leading to a distribution of bootstrapped estimates of the mean accuracy and RT for each condition. Accuracy scores were considered above chance and below ceiling if the confidence intervals did not contain .5 nor 1, which was the case across groups and experimental conditions.

Click here for additional data file.

Supplemental Information 1 Raw data

Click here for additional data file.

The authors express their gratitude to Bruno Rossion, under the supervision of which the experiments were carried out. Further thanks are directed to all participants for their cooperation, as well as Matteo Visconti di Oleggio Castello and an anonymous reviewer for their constructive comments on an earlier version of this manuscript.

Additional Information and Declarations

Competing Interests

Author Contributions

Human Ethics

Data Availability

1 Previous studies have demonstrated that increased physical similarity is associated with decreased discriminability and therefore more piecemeal processing of local cues (e.g., pixel-based intensity, or color differences; Barton et al., 2006; Orban de Xivry et al., 2008; Busigny et al., 2010; Van Belle, Lefèvre & Rossion, 2012), i.e., decreased reliance on initial holistic or global processing. Note that other authors have applied the same morphing techniques to e.g., identities differing merely in regard to a single feature (Goffaux, 2012) or a metric relation between features (Gilad-Gutnick, Yovel & Sinha, 2012).

2 For trials on which probe stimuli differed by 20% (Full 20%, see Fig. 1A), probes were selected to depict a given identity by 90% and 70% (test stimuli always depicted a given identity by 90%). For trials on which probe stimuli differed by 50% (Full 50%, Blurred 50%), the probes represented a given identity by 70% and 20%; test stimuli always depicted a given identity by 20%.

The authors declare that there are no competing interests.

Meike Ramon and Goedele Van Belle conceived and designed the experiments, performed the experiments, analyzed the data, wrote the paper, prepared figures and/or tables, reviewed drafts of the paper.

The following information was supplied relating to ethical approvals (i.e., approving body and any reference numbers):

The experiments were undertaken with the understanding and written consent of each subject, and conform to The Code of Ethics of the World Medical Association (Declaration of Helsinki). Consent for publication was obtained for individuals depicted in the figures exemplifying stimuli used.

The following information was supplied regarding data availability:

The Supplemental Information contains the raw (single trial) data for each participant and condition.

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
