# Peer review of "Real-life experience with personally familiar faces enhances discrimination based on global information"

_PeerJ, doi:10.7717/peerj.1465_

## Round 0.1 · original submission · Major Revisions

Dear Authors,

I would like you to really look at the statistical analysis of the data as suggested by peer reviewer 1. As Academic Editor for the submission I have also read the manuscript and am including in this comment for the authors my queries on the use ANOVA for categorical datasets. Please clarify this with statistical backup when you resubmit the revised manuscript as I will need to sent your revised manuscript to a statistician for verification.

Thanking you.

·

Basic reporting

Raw data: I would recommend the authors to organize the data more uniformly. The structures of the two datasets are different, and this makes it inconvenient for other researchers to load the data. It would be also better to provide the data in a non-proprietary format, such as a csv file, and provide a file describing the meaning of the different variables. Lastly, one of the files has not been anonymized: it contains subjects’ names and date of measurement (file raw_data_controls.xlsx). With respect to anonymization, please make sure that subjects are not identified by initials, but by a numeric sequence (for example, sub001, sub002, …).

Experimental design

The paradigm is very clever, and the adoption of a control group makes the findings robust to explanations based on low-level perceptual differences of the stimuli. With respect to the experimental design, the manuscript needs some minor clarifications about the paradigm and methods (an additional figure depicting the paradigm would make it clearer to the reader):
- page 7, lines 1-6: could you specify
a. if subjects had to keep fixation during the presentation of the probe stimuli (I doubt it but at the moment it is unclear);
b. the interval after which the test face appeared;
c. whether the test face was presented centrally.
- page 8, line 8: was the overall luminosity of the face pairs selected subjectively or quantified by, e.g., average pixel intensity?
- Figure 1: could the authors show an example of morphed familiar faces too?
- what was the ethnicity and gender of the stimuli used?
- page 8, line 25: was the point of subjective equality considered to be 50% for all subjects or was the psychometric function determined individually?
- page 9, line 14: were some of the same subjects enrolled in the previous experiments (Ramon 2015a, b)? Is there any chance that they already saw any version of the stimuli?

Validity of the findings

This work addresses a fundamental question about holistic vs. local processing to explain the advantage for familiar faces. I appreciated the clarity of the hypotheses and the well-controlled experimental design. However, I have major concerns about the statistical analyses.

The authors quantified the advantage for familiar stimuli using a normalized index for both reaction times and accuracies, and then conducted an ANOVA to test for significance. I am curious about why the authors adopted such an index instead of a 3x2x2 factorial design (condition, familiarity, group), because the normalized index, computed as (a-b)/(a+b), is bounded in [-1, 1] (since a, b >= 0) making the data non-normally distributed. This breaks the assumption of normality upon which the Analysis of Variance stands. Also, the computation of such an index ignores the within-subject variance of the estimate (for RTs).

Using ANOVA for categorical data (as in the case of accuracies) has been strongly criticized and shown to be less than optimal, and logit or probit mixed effect models should be preferred (see for example Jaeger, 2008; and Dixon, 2008). Generalized mixed-effects models provide a better alternative, becoming increasingly popular among researchers in our field (for a review, see Bayeen, Davidson, Bates, 2008; or Bolker, Benjamin M., et al., 2009 for a more general description, although in the field of ecology).

Importantly, these analyses can implement random effect structures for participants and stimuli, and won’t discard the within-subject variance for the different conditions (as on the other hand computing the index on average RTs for each subject and condition separately does). This will provide information about whether the effects described in this manuscript can be generalized not only across subjects, but also stimuli, resulting in more robust results.

Additional comments

This is a very good manuscript with a clear hypothesis, and a solid experimental design. As expressed above, I am concerned about the statistical analysis which are not as solid as they could be. Lastly, a recent paper by Burton, Schweinberger, Jenkins and Kaufmann (2015) provides arguments against a configural explanation for familiar face perception. In light of the hypothesis and results presented in this manuscript, it would be interesting if the authors could integrate some comments on those arguments into their discussion.

Reviewer 2 ·

Basic reporting

This paper studies the effect of real life familiarity on the recognition of differenccs in faces that were changed by morphing. The authors found that small changes that did not affect identittyincluence by prior familimiatrity. On the other hand larger differenes that did affect identity benefitted from familiarity.

Experimental design

The experimental design is rigorous.

Validity of the findings

The authors conclusions follow from the hypothesis and the experiemntal design.

---

## Round 0.2 · accepted · Accept

Dear Authors,

Thank you for all revisions made thus making it possible for the peer reviewers to accept the manuscript for publication in PeerJ.

·

Basic reporting

Thanks for addressing my points.

Experimental design

Thanks for addressing my points.

Validity of the findings

Thanks for addressing my points.

Additional comments

I thank the authors for addressing my concerns. I believe the manuscript in the current form is ready for publication.